# Diet Quality Is Associated with a High Newborn Size and Reduction in the Risk of Low Birth Weight and Small for Gestational Age in a Group of Mexican Pregnant Women: An Observational Study

**DOI:** 10.3390/nu13061853

**Published:** 2021-05-28

**Authors:** María A. Reyes-López, Carla P. González-Leyva, Ameyalli M. Rodríguez-Cano, Carolina Rodríguez-Hernández, Eloisa Colin-Ramírez, Guadalupe Estrada-Gutierrez, Cinthya G. Muñoz-Manrique, Otilia Perichart-Perera

**Affiliations:** 1Nutrition and Bioprogramming Coordination, National Institute of Perinatology Nacional de Perinatología, Montes Urales 800, Lomas de Virreyes, Mexico City 11000, Mexico; mareyeslo@unal.edu.co (M.A.R.-L.); carlapaty90@hotmail.com (C.P.G.-L.); rocameyalli@gmail.com (A.M.R.-C.); carolina_9494@hotmail.com (C.R.-H.); nutricionperinatal@gmail.com (C.G.M.-M.); 2Faculty of Medicine and Dentistry, University of Alberta, Edmonton, AB T6G 2E1, Canada; eloisa_colin@yahoo.com.mx; 3Research Direction National Institute of Perinatology, Montes Urales 800, Lomas de Virreyes, Mexico City 11000, Mexico; gpestrad@gmail.com

**Keywords:** diet quality, pregnancy, fetal programming, nutritional status, newborn

## Abstract

A high-quality diet during pregnancy may have positive effects on fetal growth and nutritional status at birth, and it may modify the risk of developing chronic diseases later in life. The aim of this study was to evaluate the association between diet quality and newborn nutritional status in a group of pregnant Mexican women. As part of the ongoing Mexican prospective cohort study, OBESO, we studied 226 healthy pregnant women. We adapted the Alternated Healthy Eating Index-2010 for pregnancy (AHEI-10P). The association between maternal diet and newborn nutritional status was investigated by multiple linear regression and logistic regression models. We applied three 24-h recalls during the second half of gestation. As the AHEI-10P score improved by 5 units, the birth weight and length increased (β = 74.8 ± 35.0 g and β = 0.3 ± 0.4 cm, respectively, *p* < 0.05). Similarly, the risk of low birth weight (LBW) and small for gestational age (SGA) decreased (OR: 0.47, 95%CI: 0.27–0.82 and OR: 0.55, 95%CI: 0.36–0.85, respectively). In women without preeclampsia and/or GDM, the risk of stunting decreased as the diet quality score increased (+5 units) (OR: 0.62, 95%IC: 0.40–0.96). A high-quality diet during pregnancy was associated with a higher newborn size and a reduced risk of LBW and SGA in this group of pregnant Mexican women.

## 1. Introduction

Nutrition during pregnancy is a key determinant of fetal growth and newborn nutritional status. The effects of intrauterine nutrition remain until later stages of life. Hediger and colleagues [1] found that children who were born underweight or small for gestational age (SGA) tended to have a higher percentage of fat mass, insulin resistance, higher blood pressure, and metabolic alterations in infancy. Likewise, low birthweight, associated with intrauterine growth restriction (IUGR), has been related to a higher incidence of cardiovascular disease and insulin non-dependent diabetes in adult life [2].

The effects of maternal diet on newborn nutritional status have been extensively studied. It is accepted that excessive exposure to glucose and fatty acids in utero may promote a higher concentration of glucose and insulin in the fetus, resulting in an accelerated growth and higher birth weight [3]. In a secondary analysis of the ROLO study, Horan et al. [4], noted that the intake of saturated fat at the end of pregnancy was positively associated with neonatal central adiposity. Similarly, a deficient consumption of cobalamin predisposes newborns to a higher adiposity [5].

Diet quality is considered as a matrix of foods and nutrients that acts synergistically and has some relationship with health [6]. During pregnancy, a high-quality dietary pattern includes a high intake of vegetables, fruits, whole grains, legumes, fish, dairy, nuts, and seeds and limits the intake of animal fat, red and processed meat, added sugars, and ultra-processed foods. It may be associated with better perinatal outcomes [7].

According to recent national data, less than half of the Mexican population reported that they usually consume vegetables and eggs. Conversely, almost 86% of the population consumes sweetened beverages on a regular basis, and one-third consumes unhealthy snacks, sweets, desserts, and ultra-processed foods [8]. Very few research exists reporting diet quality during pregnancy in Hispanic or Mexican women.

Dietary patterns may be determined a priori based on established recommendations. They may also be empirically derived based on statistical technics, with an a posteriori approach [6]. The Alternate Healthy Eating Index (AHEI) is an a priori score to evaluate diet quality and was created as an alternative to the Healthy Eating Index. AHEI includes foods and nutrients that have been associated with chronic disease risk [9]. The most recent version is the AHEI-2010, which includes vegetables, fruits, whole grains, sugar-sweetened beverages, nuts and legumes, red/processed meat, long-chain (n-3) fatty eicosapentaenoic acid (EPA) and docosahexaenoic acid (DHA), sodium, alcohol, and trans and polyunsaturated fatty acids as percentages of energy [10].

An adaptation of the original AHEI score was developed for pregnancy by Rifas and colleagues [11], (AHEI-P). It was composed of nine items that assessed vegetables, fruits, fiber, trans fatty acids, calcium, folate, and iron intake. It also included the ratio of white to red meat intake and the ratio of polyunsaturated to saturated fatty acids intake. The authors found that a higher AHEI-P score was associated with a lower risk of preeclampsia and lower blood glucose levels in healthy pregnant women. Melere and colleagues [12], re-adapted the AHEI-P for Brazilian pregnant women by adding vegetable protein (legumes) and considered calcium intake recommendations. The total score showed a positive correlation with folate, calcium, and iron intake. Ancira and colleagues [13], based on the Mexican Dietary Guidelines and international dietary recommendations, developed the Maternal Diet Quality Score (MDQS). This index includes polyunsaturated fatty acids (PUFAS), added sugars, fruits and vegetables, red meat, low-fat dairy products, legumes, and food high in saturated fat and/or added sugar. They found that a higher adherence to MDQS was associated with a reduced risk of having a low birth weight newborn [13].

While there are different scores that have been used to evaluate diet quality during pregnancy, the vast majority have been developed in high-income countries, thus limiting its applicability in Mexican pregnant women. The MDQS did not include calcium, folate, and iron intake as index items, which are essential nutrients during pregnancy. The aim of this study was to evaluate the association between maternal diet quality during the second half of pregnancy and newborn nutritional status in a group of pregnant Mexican women using a new adaptation of the AHEI-2010.

## 2. Materials and Methods

### 2.1. Study Population

This study is a secondary analysis of an ongoing Mexican prospective cohort study, OBESO (Origen bioquímico y epigenético del sobrepeso y la obesidad) (2017–2020). We included women with a single pregnancy in the first trimester of pregnancy (11 to 13 weeks of gestation), with a pre-gestational body mass index (BMI) ≥18.5 kg/m^2^, without diabetes mellitus, hypertension, chronic kidney disease, uncontrolled thyroid disease, liver diseases, or HIV. The selected women had three dietary assessments in different moments in the second trimester (described below). We also excluded those women with congenital structural malformations in their fetuses and women with chronic use of insulin, metformin, and steroids. The OBESO cohort was approved by the Committees of Ethics, Research and Biosafety of the National Institute of Perinatology (Project. No. 3300-11402-01-575-17).

### 2.2. Maternal Characteristics

The recruitment was carried out at the Maternal-Fetal Medicine Department during the first trimester visit. Trained staff explained the project, invited all women who met the criteria, and collected the informed consent. At this time, the nutritionist made the first nutrition assessment to obtain the baseline characteristics and retrospectively collect information about the pre-gestational body mass index (BMI). During the follow-up visits, which occurred every four to six weeks, we obtained the patients’ weight and completed a dietary assessment.

### 2.3. Dietary Assessment

A standardized interviewer applied a multiple-pass 24-Hour recall at 20 to 24, 24.1 to 28, 28.1 to 34, and ≥34 gestational weeks. To improve the portion size estimation, the interviewers used food replicas, as well as standard measuring cups, spoons, and glasses. Nutrient analysis was performed with the Food Processor SQL software (version 14.0, Esha Research, Salem, OR, USA). We standardized the recipes and included Mexican foods in the database. The intake of energy, macronutrients, fiber, mono, poly, saturated and trans fatty acids, cholesterol, vitamins A, C, and D, folate, calcium, iron, magnesium, selenium, and zinc was computed from three multiple-pass 24-Hour recalls. Likewise, to establish the usual energy, nutrient, and food groups intake, we considered the three dietary assessments and calculated an average for each item. Subsequently, we computed the *AHEI-10P*.

### 2.4. Alternative Healthy Eating Index-2010 for Pregnancy (AHEI-10P)

This score was created as an alternative to *AHEI-2010* for use during pregnancy [10]. The original index (AHEI-10) includes alcohol and sodium intake. Since alcohol is not recommended during pregnancy, we excluded this item. Regarding the dietary sodium intake, its assessment using dietary tools has numerous biases, and the standard method for its assessment, 24-h urine testing, is not part of the procedures of the OBESO cohort; thus, this item was also excluded [14]. Finally, we included calcium, iron, and folate intake due to their relevance during pregnancy [7]. With the exception of fish, calcium, iron, and folate, all items were scored according to the *AHEI-2010* criteria. All components were scored from 0 (worst) to 10 (best). For intermediate values, we used the equations described in Appendix A. The total *AHEI-10P* score ranged from 0 (lowest diet quality) to 120 (highest diet quality). A description and calculation for each item and scoring criteria are described in Table 1.

### 2.5. Newborn Nutritional Status

A certified dietitian obtained anthropometric measures within the first 48–72 h of birth, according to Lohman’s technique [17]. We used a Tanita WB-3000 Digital Physicians Scale to measure the weight (Tanita, Arlington Heights, IL, USA), a SECA infantometer model 207 (SECA, Hamburg, Deutschland) to measure the recumbent length, and a SECA measured tape model 212 to measure the head circumference (Hamburg, Deutschland). WHO-2006 and INTER-GROWTH-21St growth references were used for evaluating the weight for age (W/A), weight for length (W/L), length for age (L/A), body mass index for age (BMI/A), and head circumference for age (HC/A) in term and preterm newborns [18,19].

### 2.6. Potential Confounders and Intermediate Variables

Maternal age, pregestational-BMI, maternal gestational weight gain, energy intake, multivitamin use (with folic acid and iron), education level, and number of pregnancies (parity) were obtained using questionnaires that collected data on sociodemographic variables, obstetric history, and detailed information about the pregnancy.

Maternal age: This variable was dichotomized as being adolescent (<19 years) or adult (≥19 years).

Weight status: Maternal weight was measured at each visit using Lohman’s technique with a Seca 813 Digital Scale (SECA, Hamburg, Germany) [17].

Height was measured using Lohman’s technique with a Fixed Wall Stadiometer 216 for Infants and Adults (SECA, Hamburg, Germany) [17].

Pregestational BMI was classified as normal (BMI 18.5–24.9 kg/m^2^), overweight (BMI 25.0–29.9 kg/m^2^), or obese (BMI ≥ 30 kg/m^2^).

Weight gain was classified as adequate, insufficient, or excessive, according to the gestational age and pregestational-BMI, as recommended by the Institute of Medicine [20].

Total energy intake: Average energy intake was considered as Kcal/d from the 24 h recalls.

Multivitamin: The use of multivitamins was reported at each visit, and we analyzed only brands that provided folic acid and iron, which was dichotomized as use or not use. 

Education: Level of education was reported by women and was considered as low (elementary school and/or incomplete middle school), medium (completed middle school or high school), or high (technical career, bachelor’s degree and/or graduate degree).

Parity: Women were considered nulliparous (no previous pregnancy) or multiparous (one or more previous pregnancies).

Preterm birth was considered as birth at 37 weeks of gestation or less, according to the ultrasound in the first trimester; in cases where no ultrasound was available, we calculated the weeks of gestation according to the last menstrual period.

Preeclampsia was defined as increased systolic and/or diastolic blood pressure (≥140/90 mmHg), accompanied by proteinuria (≥300 mg/24 h), after 20 weeks of gestation in a previously healthy woman [21].

Gestational diabetes mellitus (GDM) was established using the one-step strategy for the oral glucose tolerance test at 24–28 weeks of gestation [22].

Newborn sex: The totality of the characteristics of reproductive structure, functions, phenotype, and genotype, differentiating the male from the female organism [23].

### 2.7. Statistical Analysis

Univariate analysis included the means and standard deviations for normally distributed variables, median and interquartile range for variables with a different distribution, and proportions for categorical variables. We used the quartile categorization of the *AHEI-10P* score for the bivariate analysis. The maternal baseline characteristics and pregnancy outcomes were described across the quartiles of *AHEI-10P*. The differences in the *AHEI-10P* scores according to the maternal characteristics, potential confounders, intermediate variables, and newborn nutritional status were evaluated using the T-student test, U-Mann Whitney test, one-way ANOVA, or Kruskal-Wallis. The chi-square test was used for categorical variables. Post hoc analyses were performed with a Bonferroni test for One-way ANOVA and with the U-Mann Whitney between pairs for non-parametric variables, and the statistical significance was adjusted in order to prevent type I errors (*p* < 0.008). In order to determine the association between diet quality and newborn weight, length, head circumference, BMI, and z-scores of the nutrition indices, we developed multiple linear regression models, including the AHEI-10P score as an independent variable. Likewise, the association between diet quality (AHEI-10P score) and low birth weight, SGA, stunting, low head circumference, being overweight, and obesity was evaluated with multiple logistic regressions models. In order to test the effect of energy intake on the relationship between diet quality and newborn nutritional status, we created an interaction term between total energy intake and diet quality; for models with a significant interaction term, this variable was reported in the results; otherwise, we report results without the interaction term in the model. We excluded preterm newborns when analyzing weight, length, head circumference, and low birth weight. Finally, we stratified the models according to the presence/absence of preeclampsia or gestational diabetes mellitus. The sample size was calculated using the difference between two independent means (birthweight) of the two different groups (high- and low-quality diets), considering a 5% probability for type I errors (*p* < 0.05) and a statistical power of 20%. The final sample was 196 women [24]. The statistical power was computed according to the effect size approach for linear multiple regression and logistic models. The analyses were performed using the statistical software package, SPSS Statistics (version 22.0, IBM, Mexico City, Mexico). The statistical significance was considered using a 95%CI and a *p* value < 0.05.

## 3. Results

### 3.1. Baseline Characteristics

We included 405 women-newborn pairs in the cohort study from January 2016 to December 2019. Women were excluded due to a lack of newborn anthropometric measures (36%, n = 148) and because the dietary assessment was not available (7.6%, n = 31). We studied 226 women-newborn pairs. Most women were adults (84.5%, n = 191). The mean age was 28.8 ± 8.1 years. The mean pregestational-BMI was 26.1 ± 5.2 kg/m^2^, 32.7% (n = 74) of which were overweight before pregnancy and 20.8% (n = 47) had obesity. In terms of sociodemographic characteristics, 68.6% (n = 155) were married or lived with a partner, 65.2% (n = 144) had a high education level, 28.5% (n = 63) a medium education level, and the remaining 6.3% (n = 14) had a low education level. Over half of the women were housewives (66.4%, n = 150), 23.9% (n = 54) were employees, and 9.7% (n = 22) were students. Regarding parity, 75.7% (n = 171) were nulliparous. 

The mean gestational weight gain in the third trimester (34.3 ± 1.7 gestational weeks) was 8.8 ± 5.2 kg, and 33.6% (n = 76) had excessive and 30.5% (n = 69) insufficient weight gain. Regarding adverse pregnancy outcomes, 8.6% (n = 19) of women had preeclampsia, 10.6% (n = 24) had GDM, 6.6% (n = 15) of newborns were preterm, and 17.3% (n = 39) had low birth weight. Table 2 presents the diet quality score and baseline characteristics according to the AHEI-10P quartiles.

Women that lived with a partner had higher diet quality scores than single women (61.7 ± 12.7 vs. 57.8 ± 12.7, *p* = 0.037). Compared to the first, second, and third quartiles, women in the highest diet quality group had the lowest frequency of preeclampsia and preterm newborns. Even though the statistical significance was obtained for GDM and low birth weight frequencies, no differences were found between the highest and lowest diet quality groups.

### 3.2. AHEI-10P and Nutrients Intake

Table 3 describes the mean energy and nutrient intake according to the AHEI-10P quartiles.

The energy intake was higher in the highest diet quartile, compared to the lowest diet quality group (2043.7 vs. 1723.5 kcal). While the macronutrient intakes were higher in the highest quartile, compared to the lowest, in proportion to the total energy, the intake of protein, lipids, and carbohydrates were no different in these groups (Table 3). When adjusted per 1000 kcal, the fiber intake was higher in the highest diet quality group, compared to the lowest diet quality group (15.2 vs. 9.5 g/d); the intake of omega 3 and 6 fatty acids was also higher in women in this diet quartile versus the lowest diet quartile (0.9 g/d vs. 0.6 and 7.2 g/d vs. 5.0 g/d, respectively, *p* < 0.01) (Table 3). Regarding the micronutrient intake, we found that the intake of vitamins A, C, and D, folate, calcium, magnesium, selenium, iron, and zinc was higher in the highest diet quality group, compared to the lowest diet quality group (*p* < 0.05) (Table 3).

### 3.3. Newborn Nutritional Status and AHEI-10P

#### 3.3.1. Nutritional Status Alterations According to AHEI-10P

Male newborns represented 49.1% (n = 111). The mean birth weight was 2846.7 ± 427.1 g, the length was 46.5 ± 2.1 cm, BMI was 13.0 ± 1.3 kg/m^2^, and head circumference was 33.7 ± 1.4 cm. The mean gestational age at birth was 38.6 ± 1.4 weeks. One third of newborns (30.5%, n = 69) were classified as SGA. Only 1.3% (n = 3) of them presented macrosomia. According to W/L, the frequency of wasted, risk of overweight, overweight, and obese newborns was 1.6% (n = 4), 16.2% (n = 30), 6.5% (n = 12), and 0.5% (n = 1), respectively. When using the BMI/A index, 1.9% (n = 4) were wasted, 5.2% (n = 11) had a risk of being overweight, and 1.4% (n = 3) were overweight. There were no cases of obesity according to BMI/A. Stunting was observed in 34.4% (n = 77) of newborns, and 5.8% (n = 13) of them had an altered head circumference growth. There were no significant differences for weight, length, head circumference, and/or BMI according to the AHEI-10P quartiles. Table 4 describes the nutritional status outcomes according to the AHEI-10P quartiles.

#### 3.3.2. Diet Quality Effect on Anthropometric Markers and Nutritional Status Alterations 

According to linear regression models, for every increase of five units in the AHEI-10P score, a higher weight, length, and W/A was observed (overall 72.70 ± 34.3 g, 0.35 ± 0.17 cm and 0.17 ± 0.07 z-score, respectively, *p* < 0.05; women without preeclampsia and/or GDM: 96.75 ± 34.71 g, 0.53 ± 0.18 cm and 0.23 ± 0.07 z-score, respectively, *p* < 0.01). In women without preeclampsia and/or GDM, the L/A increased (0.19 ± 0.09, *p* = 0.03). A trend towards a higher head circumference and HC/A was observed in newborns of women without preeclampsia and/or GDM (0.21 ± 0.12 cm, *p* = 0.07 and 0.16 ± 0.08 z-score, *p* = 0.06, respectively) (Table 5).

The risk of low birth weight decreased as the diet quality increased; for every five units of rise in the AHEI-10P score, the risk was 1.22 lower in all women and 1.27 lower in women without preeclampsia and/or GDM (*p* < 0.01). Likewise, for each five units of rise in the AHEI-10P score, the risk of SGA was 0.92 lower in all women, and it was 1.6 lower in women without preeclampsia and/or GDM (*p* < 0.01) (Table 6). Additionally, in women without preeclampsia and/or GDM, the risk of stunting was 0.6 lower for each 5 units of increase in the AHEI-10P score (*p* = 0.03) (Table 6).

Excessive maternal weight gain was associated with a higher newborn weight (β = 208 ± 64.2 g < 0.01), BMI (β = 0.70 ± 0.21 kg/m^2^, *p* < 0.01), W/A (β = 0.42 ± 0.14 z-score *p* < 0.01), W/L (β = 0.53 ± 0.21 z-score, *p* < 0.01), and BMI/A (β = 0.54 ± 0.17 z-score, *p* < 0.01). Compared with an adequate gestational weight gain, insufficient weight gain determined lower values for head circumference and HC/A (β = −0.51 ± 0.20 cm, and β = −0.35 ± 0.15 z-score, respectively, *p* < 0.05). Energy intake determined higher values of newborn W/A (β = 0.001 ± 0.00 z-score, *p* = 0.02).

Finally, multivitamin use was associated with a lower risk of newborn overweight or obesity according to BMI/A (OR: 0.14, 95%CI: 0.02–0.74).

## 4. Discussion

There are few studies evaluating diet quality during pregnancy and its association with newborn nutritional status in Latin America. As far as we know, this is the second study with this purpose in Mexico. We observed that pregnant women with higher diet quality scores (AHEI-10P) had a lower risk of low birth weight and SGA newborns and improved nutritional status markers at birth.

In the previous study conducted in Mexico, using the MDQS, the authors observed a reduced risk of LBW in the highest adherence group, compared to the lowest adherence group (OR: 0.34; 95%CI: 0.11–0.90) [13]. In other studies, that used the AHEI or pregnancy adaptations of the AHEI, Rifas and colleagues, [11], showed a lower risk of SGA in women with a high-quality diet score (OR: 0.92, 95%CI: 0.82–1.02). Similar results were found in a secondary analysis of the prospective cohort, “New Hampshire Birth Cohort Study” [25]. Rodríguez and colleagues, [24], applied an adapted version of the AHEI-2002, and they observed that birth weight and length was higher in women in the fifth diet quality quintile, compared with the lowest quintile (β = 114.1 g; 95%CI: 27.1–201.2 g and β = 0.41 cm; 95%CI: 0.03–0.80 cm). González and colleagues, [26], observed that for each unit of increase in the score of AHEI-10, the W/A z-score increased by 0.01 (95%CI: 0.002–0.02); however, when the models were adjusted, the statistical significance was lost. Similar results have been reported using other diet quality indices. In an analysis of the Australian Longitudinal Study on Women’s Health, Gresham and colleagues [27], found that, compared with women in the first quintile, those in the fifth quintile of the Australian recommended food score showed a lower risk of low birth weight (OR = 0.4; 95%CI: 0.2–0.9). In a study in two population-based mother–child cohorts in Spain and Greece, adherence to the Mediterranean diet pattern, using an a priori score, was evaluated. Women with a high adherence had a lower risk of delivering a growth-restricted newborn (RR: 0.5; 95%CI: 0.3–0.9). In smoking mothers, a higher adherence to the Mediterranean diet pattern increased weight and length at birth (Atlantic cohort β = 319 ± 124.3 g and β = 1.3 ± 0.6 cm, respectively and Mediterranean cohort β = 200 ± 81.5 g and β = 0.8 ± 0.4 cm, respectively) [28].

Our results showed that a high-quality diet was related with a greater intake of fiber, omega 3 and 6 fatty acids, vitamins A, C, and D, folate, calcium, iron, magnesium, selenium, and zinc. Dietary patterns involve a matrix of different foods that contain a number of nutrients; many of them are correlated, so it is difficult to separate their effects. However, some of these nutrients are associated with an improvement in neonatal nutrition status [29]. Fiber intake during pregnancy is important for both the mother’s health and fetal growth, and it has been associated with a higher birth weight [30]. Omega-3 long chain polyunsaturated fatty acids, in particular, have been associated with a longer gestation, higher birth weight, and less preterm birth [31]. Additionally, essential fatty acids are crucial to fetal development, particularly for cell membranes and the brain [32]. Maternal intakes of vitamins C and D and folate have been associated with higher values for length at birth. Similarly, vitamins A and D intakes have been associated with a higher head circumference in a Japanese cohort [33]. Iron supplementation appears to increase birth weight through an increase in maternal hemoglobin concentrations in the third trimester [34]. Finally, folate, vitamin A, C, and D, magnesium, selenium, and zinc have fetal programming implications that, in turn, are closely related with nutrition status at birth [2].

Besides diet quality, gestational weight gain was another factor associated with a higher newborn body mass. An excessive weight gain was related with higher values of newborn weight and BMI, and it was a protecting factor of SGA. In a population-based cohort study in the United States, Ludwing and colleagues [35], found that newborns of women who gained more than 24 kg during pregnancy were 148.9 g (95%CI: 141.7–156.0 g) heavier at birth than were infants of women who gained 8–10 kg. In a systematic review, Goldstein and colleagues, [36], found that a weight gain below IOM recommendations was related to a higher risk of SGA (OR: 1.53, 95%CI: 1.44–1.64, I2 = 82.8%).

Like excessive weight gain, energy intake was positively associated with weight and length at birth. Crume and colleagues, [37], found that newborn fat mass was increased by 4.2 g and 2.9 g for each 100 kcal from fat and carbohydrates, respectively. While the association between maternal energy intake and length at birth has been less frequently studied, Gala and colleagues, [38], found a significantly positive correlation between percentage of energy intake recommendation (RDA) and length at birth.

Another factor that was a determinant of newborn nutritional status was education level. Our results showed that women with a medium education level had a lower risk of neuro-developmental risk. In the same way, in a secondary analysis of the Generation R Study, it was found that head circumference in the first, third, and sixth month of age was lower in infants of women with a low versus those with a high education level [39]. This is important, considering that HC/A is a chronic nutritional status index, and lower education levels may be related with nutrition inequalities. Multivitamin use (including iron and folic acid) also determined a lower risk of newborn overweight or obesity. Contrary to our results, in a population-based cohort of women without GDM, Hua and colleagues, [40], observed that women that used iron and folic acid supplements were more likely to deliver a macrosomic or LGA infant (OR: 1.32, 95%CI: 1.08–1.49 and OR: 1.42, 95%CI: 1.24–1.61, respectively), as compared with women who did not take supplements. It should be noted that multivitamin use in our study was heterogeneous, and we did not control the dose, administration, duration, and/or nutrient composition.

The development and validation of a diet quality index carries some challenges. First, there is no standard reference for diet quality assessment; and second, dietary assessment has a series of biases that make it difficult to validate. We adapted the Alternate Healthy Eating Index-2010 for use during pregnancy (AHEI-10P). We believe that this version of AHEI is applicable in the Mexican population, considering the high prevalence of coronary heart disease and diabetes, and because it includes different food groups that provide different relevant nutrients in the prenatal stage [41].

In order to guarantee consistency and reduce measurement bias, interviewers were trained with a standardized methodology, and the multiple-pass version of the 24-h recall was used in three occasions to gain a closer view of the usual intake. The multiple-pass version of the 24-h recall reduces memory and portion size estimation error and may aid in providing a better food description [42]. In addition, the diet quality score was positively associated with the intake of healthy nutrients (fiber, magnesium, and folate), supporting construct validity.

As in any other dietary assessment study, heterogeneity is present. Items and cut-off points that integrate diet quality indices are not standardized, considering different food groups and different ratings systems; in addition, the dietary assessment method used can also vary (i.e., 24-h. record, food frequency). The database used for analyzing nutritional composition is another source of variability. We used the Food Processor Nutrition Analysis Software (SQL). This program uses an extensive database (including some data from Mexico) and allows for the inclusion of new foods or recipes. Finally, in the case of maternal diet quality, the moment during pregnancy in which a dietary assessment is made differs among studies.

To our knowledge, this is the first study that adapted the AHEI-10P for use in Mexican pregnant women, demonstrating that a high-quality diet is not only associated with a lower risk of chronic diseases, but also with better perinatal outcomes. This study adds to the limited literature on diet quality during pregnancy in low-income countries. The estimated effects in this study reached a statistical power greater than 80%, except for BMI, BMI/A, W/L, L/A, and HC/A. Even though more research is necessary to confirm our findings, this study shows that diet quality assessment during pregnancy could contribute to the implementation of timely nutritional strategies that may contribute to a lower incidence of low birth weight and SGA newborns.

Our study has some weaknesses that should be addressed. While we used several 24-h. recalls for estimating dietary intake, it is possible that individual and inter-individual dietary intake variations were not completely measured, and bias may therefore be an issue [43,44]. For the significant effect of diet quality on newborn L/A in the group of women without preeclampsia and/or GDM, the statistical power was low (53%). The relatively small sample size may have introduced a type II error. We did not consider physical activity, intergenesic period, or smoking habits as factors that can determine newborn nutrition status, and we did not consider anemia or pregnancy resolutions [45]. All these aspects should be considered in future studies.

## 5. Conclusions

A high-quality diet during pregnancy was associated with a higher newborn weight, length, and reduced risk of low birth weight and SGA. Women who did not develop preeclampsia and/or GDM also showed this association and had a lower risk of stunting. AHEI-10P is an alternative for evaluating diet quality in pregnant women, focusing on important nutrients for maternal and fetal health. More studies evaluating diet (quantity and quality) and its effects on newborn nutrition status in developing countries are necessary.

## Figures and Tables

**Table 1 nutrients-13-01853-t001:** The Alternate Healthy Eating Index-2010 for pregnancy (AHEI-10P) scoring method.

Component	Food Definition and Serving Size	Criteria for Minimum Score (0)	Criteria for Maximum Score (10)
Vegetables, servings/d ^1^	Any type of vegetable in any preparation. One serving = 1 raw cup or ½ cup cooked, high in HCO (1 cup = 236.59 g). Does not include potato, corn, or avocado	0	≥5
Fruit, servings/d ^1^	Any natural and whole fruit (not fruit juice). One Serving = according to the Mexican food exchange system [15].	0	≥4
Whole grains, g/d ^1^	Whole grains and whole grain non-refined cereals were considered (corn tortilla, pozole corn, popcorn, oats, amaranth, brown rice and pasta, and granola). One serving = Serving containing 15 g of HCO, according to the Mexican food exchange system [15].	0	75 ^2^
Sugar-sweetened beverages and fruit juice, servings/d ^1^	Includes any industrialized juice or natural juice, soft drinks, or flavored water powder. One serving = 240 mL. This does not include coffee or tea with sugar or flavored waters due to its variable sweetener content. Besides, it is not equal to the sugar content of other industrialized beverages.	≥1	0
Nuts and legumes, servings/d ^1^	Legumes include different types of beans, lentils, and chickpeas. Nuts include walnuts, almonds, pistachios, peanuts, pine nuts, sunflower seeds, and peanut butter seeds. One serving of legumes = ½ cup. One serving of oilseeds and seeds = 1.5 tablespoons, 28 g, or 15 mL.	0	≥1
Red/processed meat, servings/d ^1^	Processed meat refers to meats that have undergone a transformation process through salting, curing, fermentation, or smoking. Red meats include beef, lamb, pork, or beef and poultry viscera. One serving of red meat = 113.4 g and processed meat = 42.5 g.	≥1.5	0
–Trans fat, % of energy ^1^	The amount that trans fatty acids contribute to TCV in percentage.	≥4	≤0.5
Fish, servings/d	Fish is considered the main source of EPA and DHA fatty acids, so its assessment is comparable to the direct evaluation of EPA and DHA intake. This category does not include seafood. The suggested amount of fish intake was adapted to an AND of 250 g per week during pregnancy. One serving = 35.7 g/d (250 g / 7 = 35.7 g).	0	≥35.7 g
PUFA, % of energy ^1^	The amount that polyunsaturated fatty acids contribute to TCV.	≤2	≥10
Dietary calcium intake, mg/d	Calcium is needed for bone formation, fetal growth, and development. A low calcium intake is implicated in hypertensive disorders. An adequate intake during pregnancy is important for optimizing perinatal outcomes [16]. We established an average calcium intake through a serial dietary assessment (mg/d).	0	≥1000
Dietary iron intake, mg/d	Due to hematologic changes and increased needs during pregnancy, iron is essential. A lack of iron leads to anemia and affects physical working capacity, brain function, and behavior. Iron deficiency increases the risk of adverse perinatal outcomes. In low-resource settings, iron-deficiency anemia is prevalent and is often exacerbated by infectious diseases [7]. We established an average iron intake through a serial dietary assessment (mg/d).	0	≥28
Dietary folate intake, mcg/d	Folate is critical for normal fetal development. Folate insufficiency before pregnancy is a proven risk factor for the development of NTDs and other congenital malformations. Additionally, folate is important in women for the prevention of macrocytic anemia and is implicated in maintaining cardiovascular health and cognitive function [7]. We established an average folate intake through a serial dietary assessment (mcg/d of DFE).	0	≥750

^1^ Included in the original Alternate Healthy Eating Index (AHEI-2010). ^2^ Amount established as ideal per woman according to the original score. TCV: Total caloric value. EPA: Eicosapentaenoic acid. DHA: Docosahexaenoic acid. AND: Academy of Nutrition and Dietetics. NTDs: Neural Tube Defects. DFE: Dietary folate equivalents.

**Table 2 nutrients-13-01853-t002:** Diet quality score, baseline characteristics, and maternal perinatal outcomes according to the *AHEI-10P* quartiles.

			AHEI-10P Quartiles n (%)				
	Diet Quality Score AHEI-10P X ± DE	P^a^	First (n = 56)	Second (n = 59)	Third (n = 54)	Fourth (n = 57)	P^b^
Maternal age							
Adults	60.7 ± 12.5	0.46	47 (83.9%)	46 (78.0%)	47 (87.0%)	51 (89.5%)	0.35
Adolescents	59.0 ± 12.6		9 (16.1%)	13 (22.0%)	7 (12.0%)	6 (10.5%)	
Pregestational status							
Normal	60.3 ± 13.1	0.67	25 (44.6%)	27 (45.8%)	25 (46.3%)	28 (49.1%)	0.48
Overweight	61.4 ± 11.5		14 (25.0%)	23 (39.0%)	19 (35.2%)	18 (31.6%)	
Obesity	59.2 ± 14.2		17 (30.4%)	9 (15.3%)	10 (18.5%)	11 (19.3%)	
Parity							
Multiparous	60.2 ± 13.2	0.86	17 (30.4%)	13 (22.0%)	10 (18.5%)	15 (26.3%)	0.49
Nulliparous	60.5 ± 12.7		39 (69.6%)	46 (78.0%)	44 (81.5%)	42 (73.7%)	
Education level							
Low	63.4 ± 12.6	0.56	3 (5.6%)	5 (8.6%)	2 (3.8%)	4 (7.1%)	0.94
Medium	61.2 ± 13.1		37 (68.5%)	36 (62.1%)	36 (67.9%)	35 (62.5%)	
High	63.0 ± 13.7		14 (25.9%)	17 (29.3%)	15 (28.3%)	17 (30.4%)	
Civil status							
Single	57.8 ± 12.7	0.03	20 (35.7%)	22 (37.3%)	17 (31.5%)	12 (21.1%)	0.23
Married/consensual union	61.7 ± 12.7		36 (64.3%)	37 (62.7%)	37 (68.5%)	45 (78.9%)	
Occupation							
Housewife	60.2 ± 12.2	0.85	39 (69.6%)	34 (57.6%)	43 (79.6%)	34 (59.6%)	0.18
Employed	60.5 ± 14.4		13 (23.2%)	16 (27.1%)	8 (14.8%)	17 (29.8%)	
Student	61.8 ± 13.4		4 (7.1%)	9 (15.3%)	3 (5.6%)	6 (10.5%)	
Multivitamin use							
Used	61.2 ± 13.0	0.17	40 (71.4%)	44 (75.9%)	37 (68.5%)	46 (80.7%)	0.47
Not used	58.5 ± 12.1		16 (28.6%)	14 (24.1%)	17 (31.5%)	11 (19.3%)	
Calcium supplementation							
Used	64.8 ± 8.3	0.21	1 (1.8%)	3 (5.2%)	3 (5.6%)	6 (10.5%)	0.25
Not used	60.2 ± 13.0		55 (98.2%)	55 (94.8%)	51 (94.4%)	51 (89.5%)	
Gestational weight gain							
Adequate	59.7 ± 12.2	0.41	20 (35.7%)	25 (42.4%)	19 (35.2%)	17 (29.8%)	0.51
Insufficient	62.1 ± 12.1		14 (25.0%)	14 (23.7%)	19 (35.2%)	22 (38.6%)	
Excessive	59.6 ± 14.1		22 (39.3%)	20 (33.9%)	16 (29.6%)	18 (31.6%)	
Preeclampsia							
Present	57.9 ± 8.2	0.19	4 (7.1%)	5 (8.5%)	9 (16.7%)	1 (1.8%)	0.04
Not present	60.7 ± 13.2		52 (92.9%)	54 (91.5%)	45 (83.3%)	56 (98.2%)	
GDM							
Present	63.2 ± 10.0	0.26	1 (1.8%)	7 (11.9%)	11 (20.4%)	5 (8.8%)	0.01
Not present	60.2 ± 13.1		55 (98.2%)	52 (88.1%)	43 (79.6%)	52 (91.2%)	
Preterm birth							
Present	58.1 ± 13.5	0.46	4 (7.1%)	2 (3.5%)	8 (14.8%)	1 (1.8%)	0.03
Not present	60.7 ± 12.8		52 (92.9%)	57 (96.6%)	46 (85.2%)	56 (98.2%)	
Low birth weight							
Present	60.0 ± 13.9	0.82	12 (21.4%)	5 (8.5%)	14 (25.9%)	8 (14.0%)	0.06
Not present	60.6 ± 12.7		44 (78.6%)	54 (91.5%)	40 (74.1%)	49 (86.0%)	

The statistical significance (*p* < 0.05) was tested with the T-student or ANOVA (P^a^) test for the means and chi-square or Fisher’s exact tests (P^b^) for the frequencies. GDM: gestational diabetes mellitus.

**Table 3 nutrients-13-01853-t003:** Total energy and nutrient intake during the 2nd half of pregnancy according to the AHEI-10P quartiles.

Energy and Nutrients	Total (n = 226)	AHEI-10P Quartiles				^Ϯ^ P
		First (n = 56)	Second (n = 59)	Third (n = 54)	Fourth (n = 57)	
Energy (kcal/d)	1813.2 (1499.4–2183.8)	1723.5 (1443.6–1954.7)	1795.9 (1392.7–2085.7)	1850.9 (1484.4–2270.1)	2043.7 (1623.8–2465.8)	0.007
Protein (%TCV)	17.2 (15.1–19.3)	17.4 (15.2–19.3)	17.4 (15.4–19.5)	16.6 (14.6–20.7)	17.1 (15.2–18.7)	0.692
Protein (g/d)	80.2 ± 23.3	74.5 ± 22.8	78.1 ± 21.5	81.5 ± 23.7	86.6 ± 23.9	0.007
Carbohydrate (%TCV)	53.4 ± 6.3	52.9 ± 6.7	53.1 ± 5.9	53.3 ± 5.7	54.3 ± 6.7	0.272
Carbohydrate (g/d)	233.3 (195.4–294.6)	218.7 (172.5–274.2)	229.0 (188.9–282.2)	238.6 (191.1–314.8)	255.7 (225.2–337.8)	0.001
Fat (%TCV)	30.2 ± 5.2	30.1 ± 4.9	30.3 ± 5.2	30.4 ± 5.0	30.0 ± 5.7	0.950
Fat (g/d)	60.4 (46.9–75.8)	54.1 (42.5–71.1)	58.9 (43.0–72.2)	64.1 (48.8–82.8)	64.1 (47.8–85.4)	0.023
Fiber g/1000 kcal	11.6 (9.2–15.2)	9.5 (7.6–11.4)	11.1 (8.9–13.8)	12.7 (10.5–15.2)	15.2 (10.5–17.7)	0.000
Fiber (g/d)	21.8 (16.4–26.9)	15.1 (12.7–19.6)	19.3 (15.5–24.1)	24.9 (20.0–27.7)	27.3 (22.4–34.8)	0.000
Saturated fatty acids (%TCV)	9.2 ± 2.2	9.3 ± 2.3	9.7 ± 2.3	9.3 ± 2.1	8.6 ± 1.8	0.079
Monounsaturated fatty acids (%TCV)	9.5 ± 2.3	9.2 ± 2.0	9.5 ± 2.4	9.3 ± 2.0	9.5 ± 2.5	0.518
W-3 fatty acids (g/d)	0.7 (0.5–1.0)	0.6 (0.4–0.9)	0.6 (0.5–0.9)	0.7 (0.5-1.0)	0.9 (0.6–1.2)	0.000
W-6 fatty acids (g/d)	5.8 (4.4–7.7)	5.0 (4.1–6.7)	5.8 (4.3–7.5)	5.8 (4.3–7.2)	7.2 (5.0–9.2)	0.001
Cholesterol (mg/d)	260.6 (191.6–356.9)	261.2 (174.8–345.7)	261.7 (202.1–373.2)	234.5 (193.9–346.3)	270.5 (201.4–340.0)	0.673
Vitamin A (UI/d)	5879.3 (3449.1–11459.4)	3805.3 (2688.2– 6714.0)	4687.3 (3018.9–8338.0)	8759.8 (3745.2–15595.5)	9491.6 (5862.9–15184.5)	0.006
Vitamin C (mg/d)	114.1 (69.5–181.8)	85.7 (51.9–131.0)	96.5 (55.3–160.9)	142.3 (75.7–192.5)	166.0 (91.4–221.5)	0.000
Folate (mcg/d)	293.6 (223.2–404.0)	224.8 (169.9–293.4)	258.8 (207.7–326.2)	318.2 (273.6–414.3)	419.5 (292.7–568.3)	0.000
Vitamin D (UI/d)	139.6 (81.4–200.7)	114.5 (56.5–187.6)	119.6 (69.7–191.4)	144.3 (95.5–214.8)	185.3 (112.2–209.4)	0.006
Calcium (mg)	877.2 (685.7–1074.5)	765.4 (523.3–991.5)	780.8 (641.3–1048.1)	950.0 (718.1–1167.8)	983.8 (779.4–1140.0)	0.000
Iron (mg)	11.5 (9.4–15.1)	10.7 (7.2–12.8)	11.1 (8.6–12.7)	11.7 (10.0–14.9)	15.1 (11.2–18.1)	0.000
Magnesium (mg/d)	279.7 (230.7–341.9)	230.9 (183.9–265.4)	255.4 (224.6–312.4)	294.4 (257.5–362.4)	363.2 (285.2–417.5)	0.000
Selenium (mcg/d)	74.4 (60.1–90.7)	70.8 (53.4–87.0)	75.6 (61.9–88.1)	73.0 (59.0–90.0)	79.5 (65.7–103.4)	0.012
Zinc (mg/d)	9.3 (7.3–11.9)	8.7 (6.7–10.8)	8.7 (6.6–11.2)	9.9 (7.7–11.3)	10.2 (8.2–13.0)	0.006

The values are the medians and interquartile range or means ± SDs. TCV: Total caloric value. ^Ϯ^ The statistical significance was tested between the first and fourth quartiles with the T-Student or U Mann-Whitney test.

**Table 4 nutrients-13-01853-t004:** Newborn nutritional status according to the *AHEI-10P* quartiles.

Alterations of Newborn Nutritional Status	Total N (%)	Diet Quality Score AHEI-10P X ± DE ^a^	*p*	*AHEI-10P* Quartiles ^b^				*p*
				First Quartile (n = 56)	Second Quartile (n = 59)	Third Quartile (n = 54)	Fourth Quartile (n = 57)	
Small for gestational age								
Present	69 (30.5%)	61.2 ± 14.1	0.60	21 (37.5%)	11 (18.6%)	17 (31.5%)	20 (35.1%)	0.12
Not present	157 (69.5%)	60.2 ± 12.3		35 (62.5%)	48 (81.4%)	37 (68.5%)	37 (64.9%)	
Stunted (L/A) ^1^								
Present	77 (34.4%)	62.0 ± 13.9	0.19	19 (33.9%)	16 (27.6%)	17 (32.1%)	25 (43.9%)	0.31
Not present	147 (65.6%)	59.7 ± 12.3		37 (66.1%)	42 (72.4%)	36 (67.9%)	32 (56.1%)	
Altered head circumference growth								
Present	13 (5.8%)	66.3 ± 10.2	0.08	2 (3.6%)	1 (1.7%)	5 (9.4%)	5 (8.9%)	0.21
Not present	210 (94.2%)	60.0 ± 13.0		54 (96.4%)	57 (98.3%)	48 (90.6%)	51 (91.9%)	
Overweight (W/L)								
Present	12 (6.5%)	66.3 ± 10.2	0.27	2 (4.3%)	2 (3.8%)	4 (9.3%)	4 (9.3%)	0.55
Not present	173 (93.5%)	60.0 ± 13.0		45 (95.7%)	50 (96.2%)	39 (90.7%)	39 (90.7%)	
Overweight (BMI/A)								
Present	3 (1.4%)	60.3 ± 12.8	0.96	1 (1.9%)	-	1 (2.2%)	1 (1.8%)	0.76
Not present	209 (98.6%)	60.8 ± 13.0		51 (98.1%)	57 (100%)	45 (97.8%)	56 (98.2%)	
Obesity (W/L)								
Present	1 (0.5%)	53.9	0.64	-	1 (1.9%)	-	-	0.46
Not present	184 (99.5%)	59.8 ± 12.7		47 (100%)	51 (98.1%)	43 (100%)	43 (100%)	
Macrosomia								
Present	3 (1.3%)	53.7 ± 8.3	0.33	2 (3.6%)	-	1 (1.9%)	-	0.28
Not present	223 (98.6%)	60.5 ± 12.8		54 (96.4%)	59 (100%)	53 (98.1%)	57 (100%)	

The statistical significance (*p* < 0.05) was tested with the T-Student test. ^a^ Chi-square, or Fisher’s exact tests. ^b^ W/A: weight for age. L/A: length for age. HC/A: head circumference for age W/L: weight for length. BMI/A: body mass index per age.

**Table 5 nutrients-13-01853-t005:** Effect of diet quality on anthropometric markers.

	AHEI-10P ^1^							
Anthropometric Markers	B	EE	B Std	*p*	95%CI		R^2^	P ^¥^
**Overall (n = 226)**								
^Ϯ^ Weight (g)^2^	72.70	34.29	0.48	0.03	5.07	140.34	0.13	0.00
^Ϯ^ Lenght (cm)^2^	0.35	0.17	0.47	0.04	0.01	0.70	0.10	0.00
BMI (kg/m^2^)	0.03	0.03	0.06	0.33	−0.03	0.10	0.05	0.02
Head circumference (cm)^2^	−0.01	0.03	−0.02	0.75	−0.07	0.05	0.13	0.00
^Ϯ^ z-score W/A	0.17	0.07	0.53	0.01	0.03	0.32	0.08	0.00
z-score W/L	0.00	0.03	0.01	0.87	−0.06	0.07	0.003	0.40
z-score L/A	0.00	0.02	0.00	0.99	−0.05	0.05	0.02	0.18
z-score BMI/A	0.01	0.02	0.04	0.57	−0.03	0.06	0.04	0.05
z-score HC/A	−0.00	0.02	−0.02	0.73	−0.05	0.04	0.04	0.03
Women without preeclampsia or GDM (n = 190)								
^Ϯ^ Weight (g) ^2^	96.75	34.71	0.70	0.00	28.21	165.29	0.16	0.00
^Ϯ^ Length (cm) ^2^	0.52	0.18	0.71	0.00	0.15	0.90	0.15	0.00
BMI (kg/m^2^)	0.01	0.03	0.04	0.57	−0.04	0.08	0.07	0.01
^Ϯ^ Head circumference (cm) ^2^	0.21	0.12	0.47	0.07	−0.02	0.44	0.10	0.00
^Ϯ^ z-score W/A	0.23	0.07	0.75	0.00	0.07	0.38	0.12	0.00
z-score W/L	−0.01	0.03	−0.03	0.69	−0.07	0.05	−0.002	0.47
^Ϯ^ z-score L/A	0.19	0.09	0.54	0.03	0.01	0.38	0.05	0.03
z-score BMI/A	0.00	0.02	0.00	0.93	−0.05	0.05	0.04	0.07
^Ϯ^ z-score HC/A	0.16	0.08	0.49	0.06	−0.00	0.33	0.04	0.07

Multiple Linear Regression Models. Models adjusted by = maternal age (years), pregestational-BMI (reference = normal weight), maternal weight gain (reference = adequate), energy intake (kcal/d), multivitamin use (reference = No use), education level (reference = low), parity (reference = multiparous women), and sex (reference = girls). ^1^ Expressed as each five units of the original AHEI-10P score. ^2^ Preterm newborns excluded. B = Regression coefficients. EE = Estandard error. B std = Estandarized regression coefficients. P = *p* value. 95%CI = 95% confidence interval. R^2^ = adjusted R-squared. P ^¥^ = Model significance. ^Ϯ^ Models for which the interaction term between diet quality and energy intake was significant (<0.05). AHEI-10P = Alternate healthy eating index for pregnancy, 2010. BMI = Body mass index. W/A = Weight for age. W/L = Weight for length. L/A = Length for age. BMI/A = Body mass index for age. HC/A = Head circumference for age. GDM: gestational diabetes mellitus.

**Table 6 nutrients-13-01853-t006:** Effect of diet quality on nutritional status alterations.

		AHEI-10P ^1^						
	Nutritional Status Alterations	B	OR	95%CI		*p*	R^2^	*p* ¥
Overall (n = 226)	^Ϯ^ Low birth weight ^2^	−0.79	0.45	0.25	0.79	0.00	0.17	0.07
	^Ϯ^ SGA	−0.63	0.52	0.34	0.82	0.00	0.18	0.00
	^Ϯ^ Stunting	−0.32	0.72	0.48	1.06	0.10	0.09	0.32
	Altered head circumference	0.19	1.20	0.94	1.54	0.13	0.18	0.25
	Overweight and obesity (W/L)	0.12	1.13	0.89	1.44	0.29	0.25	0.05
	Overweight and obesity (BMI/A)	0.20	1.22	0.88	1.69	0.22	0.46	0.00
Women without preeclampsia or GDM (n = 190)	^Ϯ^ Low birth weight ^2^	−0.82	0.44	0.24	0.79	0.00	0.20	0.09
	^Ϯ^ SGA	−0.96	0.38	0.22	0.64	0.00	0.27	0.00
	^Ϯ^ Stunting	−0.47	0.62	0.40	0.97	0.03	0.14	0.09
	Altered head circumference	0.15	1.16	0.88	1.54	0.28	0.17	0.51
	Overweight and obesity (W/L)	0.00	1.00	0.83	1.21	0.96	0.28	0.02
	Overweight and obesity (BMI/A)	0.14	1.15	0.82	1.59	0.40	0.47	0.00

Logistic regression models. Models adjusted by = maternal age (years), total energy intake (kcal/d), pregestational-BMI category (reference = normal weight), maternal weight gain category (reference = adequate), multivitamin use (reference =No Use), educational level (reference = low), parity (reference = multiparous women), and sex (reference = girls). ^1^ Expressed as each five units of the original AHEI-10P score. ^2^ Preterm newborns excluded. B = Regression coefficients. OR = Odds ratio. 95%CI = 95% confidence interval. P = *p* value. R^2^ = Nagelkerke R-squared. *p* **^¥^** = Model significance. ^Ϯ^ Models for which the interaction term between diet quality and energy intake was significant (<0.05). AHEI-10P = Alternate healthy eating index for pregnancy, 2010. SGA: small for gestational age. W/L = Weight for length. BMI/A = Body mass index for age. GDM: gestational diabetes mellitus. W/L: weight for length.

## Data Availability

Not applicable.

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
