# Peer review of "Diet Quality Is Associated with a High Newborn Size and Reduction in the Risk of Low Birth Weight and Small for Gestational Age in a Group of Mexican Pregnant Women: An Observational Study"

_nutrients, 2021, doi:10.3390/nu13061853_

Round 1

Reviewer 1 Report

This is a well written, interesting manuscript, that will contribute to the literature.

Author Response

Thanks for your comments. We would like to comment you that we made four adjustments to the paper:

1.- We considered that newborn sex is an important determinant of anthropometric markers (lines 174-175); therefore, it is a covariate that is usually included in multivariable adjusted models of maternal diet quality studies. We re-run all the multivariable models with results quite similars (Tables 5 and 6). We reformulated the results in lines 283-291, 303-309 and 320-338.

Also, we adjusted de statistical parameters in tables 5 and 6.

2.- We adjust the cut-off point for folate and iron intake, according to the Mexican dietary intake recommendations (750 mcg/d and 28 mg/d, respectively) (Table 1). This resulted in a minimal change in the AHEI-10P distribution, with minor differences in the size of each quartile. Everything is updated in the current manuscript (tables 2, 3 and 4). We rewrote two paragraphs in order to the minor changes (lines 228-234 and 242-252). The esencial information is the same.

3.-In the baseline characteristics of results section we maded a mistake regarding socioeconomics features. We wrote “socioeconomic status” instead of education level, but throughout the whole document and from early stages of writing we always used the education level (lines 214-215) (Table 2).

  1. -We rewrote some phrases throught the manuscript:

- Lines 9-14, 136, 240, 265, 274, 352, 361, 452-459, 480-482.

  • Paragraph 320-338 was rewritten to summarize the results and only indicate the covariates that had an independet statistical significance on the newborn nutrition status, without stratifying by presence or absence of preeclampsia and/or GDM.

Unfortunately I was wrong to send the corrections to reviewer 3, so I send them to you, I hope it does not cause major inconvenience.

Reviewer 2 Report

The authors evaluated the association between maternal diet quality during 2nd half pregnancy and newborn nutritional status in a group of pregnant women. Main results report that pregnant women with higher diet quality scores had a lower risk of low birth weight and SGA newborns and improved nutritional status markers at birth.

The authors stated they focused on diet evaluation in the 2nd half of pregnancy, but women were recruited between 11-13 gestation weeks. One can assume diet was evaluated at some point between recruitment and 20 gestation weeks. Did they have dietary data during the first half of pregnancy? Alcohol consumption was excluded from the diet quality index, but alcohol consumption can still be present to some extent during the first part of pregnancy.

In the manuscript, it is unclear if the authors previously validated the AHEI-10P in a different sample of pregnant women.

The sample size calculation was not included in the manuscript. Please add how the sample size or study power was calculated.

There were no underweight women before pregnancy?

In methods, why you dichotomized women’s age?

In Table 3 authors show estimated energy intake in their sample. Looking at the first and second AHEI-10P quartiles is evident a very low energy intake, with p25 being ~1300 Kcal/day. This fact indicates a deficient energy intake in some women mainly due to dietary survey bias?

The authors did discuss results against previous literature. Still, they did not confer on potential mechanisms that may explain some of the associations between diet quality & newborn nutritional outcomes.

I believe the title should be tone down considering the observational nature of the study.

Authors must carefully review the English language.

Minors:

-           Lines 30, 33, 64, 73, 384/385: add citations

-           Table 2: what Pa and Pb stand for?

-           Table 3: similar letters confuse the significant contrasts between AHEI-10P quartiles

-           Limitations should include the use of a dietary survey to measure energy intake and diet's characteristics

Reviewer 3 Report

This research is interesting because the author tried to differentiate the quality from the quantity of the maternal nutrition on fetal growth as well as neurodevelopmental risk.

However, from practical obstetrical point of view, there is a couple of questions. For example, a thin woman needs to take more calories to obtain adequate fetal growth. On the other hand, obese woman tends to bear heavy for date infant even if she eats less. Looking at the subjects in this study , more than half falls into overweight and obesity in which weight gain should rather be restricted. Nonetheless, high diet quality seems to be linked with higher Cal as well as a higher newborn size.

As mentioned in this manuscript, the recommendation to the pregnant women about the expected maternal weight gain during pregnancy is based on the pregestational BMI. According to the Institute of Medicine guidelines, there are four divisions, i.e, underweight (< 18.5), normal (18.5-25.0), overweight (25-30), and obesity (>30). For example, underweight women should gain more weight compared to the other groups. On the other hand, obese women should gain less weight. In other words, individual nutritional intervention is important.

Q1. The author should also consider underweight women by dividing four groups, including underweight women.

Q2. Since recommended weight gain differs extensively from underweight to obese women, analysis on weight gain in Table 2, should be expressed in detail such that the four subdivision of the pregestational maternal nutrition is considered.

Q3. Since smoking is one of the important factors in relation to the fetal growth, numbers of smokers should be mentioned.

Q4. As for the assessment or index of neurodevelopmental risk, is HC/A a reliable marker? If so, the author should explain in detail how this index is reliable. 

Round 2

Reviewer 2 Report

The authors have addressed the comments made to the manuscript in the previous revision round. Importantly, relevant methodological aspects are now included, and the Results and Discussion sections are now improved.

However, this report's reliance on a dietary survey is still not explicitly described as one limitation of the study (i.e., in between lines 444-449). Several old and recent reports in the literature have highlighted this issue consistently. Please include this issue as a limitation of your study. Authors may also include one (or several) citations to support this criticism.

Author Response

We agree. We add this issue as a limitation (lines 444-446) and support it with two citations

Reviewer 3 Report

I cannot find a comment of smoking in the Discussion section (line 429). Please make sure. 

Author Response

You are right. The correct line is 450 in the last paragraph just before conclusions (In the file named nutrients-1156710)
